# Effects of education and age on the experience of youth violence in a very low-resource setting: a fixed-effects analysis in rural Burkina Faso

Naasegnibe Kuunibe [1], Mamadou Bountogo,[2,3] Lucienne Ouermi,[2] Ali Sié,[2] Till Bärnighausen,[4,5,6,7] Guy Harling [5,6,8,9]

For numbered affiliations see end of article.

**Correspondence to**
Dr Guy Harling;
g.harling@ucl.ac.uk

## ABSTRACT

**Objective** The study aimed to investigate the effects of education and age on the experience of youth violence in low-income and middle-income country settings.

**Design** Using a standardised questionnaire, our study collected two waves of longitudinal data on sociodemographics, health practices, health outcomes and risk factors. The panel fixed-effects ordinary least squares regression models were used for the analysis.

**Settings** The study was conducted in 59 villages and the town of Nouna with a population of about 100 000 individuals, 1 hospital and 13 primary health centres in Burkina Faso.

**Participants** We interviewed 1644 adolescents in 2017 and 1291 respondents in 2018 who participated in both rounds.

**Outcome and exposure measures** We examined the experience of physical attacks in the past 12 months and bullying in the past 30 days. Our exposures were completed years of age and educational attainment.

**Results** A substantial minority of respondents experienced violence in both waves (24.1% bullying and 12.2% physical attack), with males experiencing more violence. Bullying was positively associated with more education (β=0.12; 95% CI 0.02 to 0.22) and non-significantly with older age. Both effects were stronger in males than females, although the gender differences were not significant. Physical attacks fell with increasing age (β=−0.18; 95% CI −0.31 to −0.05) and this association was again stronger in males than females; education and physical attacks were not substantively associated.

**Conclusions** Bullying and physical attacks are common for rural adolescent Burkinabe. The age patterns found suggest that, particularly for males, there is a need to target violence prevention at younger ages and bullying prevention at slightly older ones, particularly for those remaining in school. Nevertheless, a fuller understanding of the mechanisms behind our findings is needed to design effective interventions to protect youth in low-income settings from violence.

## INTRODUCTION

Violence globally represents both public health and an economic problem. In health terms, violence generates both mortality and morbidity. Violence is estimated to cause 1.3 million deaths annually, accounting for 2.5% of global mortality.[1] Violence often requires acute health service access (eg, assault victims requiring emergency hospital care) and can result in long-term physical disability, depression or reproductive health problems.[1,2] Violence also affects local economies through workforce absenteeism, loss of productivity and loss of human capital. Families can fall into poverty if a breadwinner dies or becomes permanently disabled due to violence.[1,3]

Youth violence—which includes bullying, physical fighting, sexual and physical assault and homicide—is particularly problematic because it generates higher economic, welfare and criminal justice costs.[4] In addition to death, injury and psychological harm, youth violence can lead to increased subsequent health risks behaviours such as smoking, substance abuse, unsafe sex and further violence.[5,6] An estimated 200 000 youth homicides occur each year, 83% among males, nearly all in low-income and middle-income

countries (LMICs), with particularly high rates in Latin America and the Caribbean (LAC) and sub-Saharan Africa.[4] In sub-Saharan Africa in particular, historical, economic and social factors continue to expose youth to violence.[7 8] Although violence affects all youth, adolescent girls and young women (AGYW) in LMICs appear to be most affected.[9–11] Social and cultural norms, such as arranged and teenage marriage and denial of resource access to women, often keep AGYW economically dependent on males and thus vulnerable to abuse, especially from intimate partners.[12–14] AGYW experience all forms of violence (emotional, physical and sexual), perpetrated by men and often other females in domestic and social settings.[15–18]

Many factors have been proposed as determinants of youth violence receipt and perpetration in LMICs, often adopting or adapting Heise's integrated ecological framework for violence against women.[19 20] This framework conceptualises violence as a multifaceted phenomenon grounded in an interplay among personal, situational and sociocultural factors. We modified this conceptualisation to cover all forms of adolescent violence, concentrating on individual-level and microsystem (family/household/relationship) factors given our focus within a specific geography. We focus on two individual-level factors believed to play an important role in determining violence experience: age and education. Given the magnitude of youth violence, evidence of associations between violence, age and educational attainment, and calls for youth violence interventions,[1 2] causal evidence on whether policies based on age or education are likely to affect violence levels is important. Past longitudinal analyses of adolescent violence often focused on the consequences of violence experience, rather than predictors of violence itself.[21 22] We instead focus on predictors of adolescent violence experience, to contribute to the upstream prevention of violence, rather than efforts to break the connection between violence and its sequelae.

Age is strongly associated with both the experience and perpetuation of violence.[3 23] Youth aged 15–29 are more likely to both experience and perpetrate violence than older adults, often experiencing violence perpetrated by their older peers or family members.[3 23] Males are more likely to perpetrate violence, while young girls and women are more likely to experience it.[9 10 14 24 25] However, evidence on the causal effect of age on violence is comparatively scarce, with few even cross-sectional studies explicitly focused on youth.[3 11 26]

Education is theorised to protect against violence, since more-educated persons are less likely to either perpetrate or experience violence.[26 27] Evidence shows that women without education were 5.6 times more likely than those with college education to experience intimate partner violence (IPV). Similarly, wives of uneducated men were 1.84 times more likely than those whose husbands had college education to experience IPV. Even at the community level, the likelihood of IPV declined as community male and female literacy increased (after controlling for individual level factors).[28] However, causal evidence on the effect of education on youth violence in sub-Saharan Africa is again limited. Two studies have used changes in national school policy as natural experiments in this context. One focused on violence, finding that a 1-year increase in grade attainment was associated to a nine percentage-point reduction in the probability of experiencing sexual violence in Uganda, but no significant effect in Malawi.[27] A second focused on sexual health, finding an additional year of schooling was associated with 0.11 fewer births and 14 percentage points less teen marriage in Ghana.[29] Both of these studies necessarily assess the overall impact of policy change, rather than the increase in education alone, and it is unclear how their findings extrapolate to lower educational attainment settings.

Burkina Faso is a landlocked country in West Africa, which despite economic and political reforms remains one of the poorest in the world, with about half of its population living below the international poverty line.[30] Economic deprivation is strongly associated with youth violence.[7 23 31] The country is very young, with around 45% aged under fifteen and a further 20% aged 15–24 in 2015.[30 32] Educational access and youth literacy are limited, with only 13% of adults having completed primary education.[32 33] Within Burkina Faso, poverty is highest in the Boucle du Mouhoun region.[34] Violence experience is common for young Burkinabe, with lifetime physical violence prevalence reported at 47%–80% and sexual violence at 33%–51%.[35 36] Adverse psychological and mental health outcomes commonly follow such experiences.[37 38] However, studies of Burkinabe youth violence have generally used cross-sectional designs and have not explored the effects of education or age specifically.[3 39]

We, therefore, analysed longitudinal data on adolescents in Boucle de Mouhoun to assess the effects of age and education on violence experience. The potential for educational interventions to have violence-specific benefits in such high-poverty, low-education settings is likely to be substantial.[37 38] By using fixed-effects analysis, we were able to exclude time-specific and time-invariant confounders, something particularly important given the many unobserved predictors of violence perpetration and victimisation.[22 40]

## METHODS
### Study design
We used data from the Nouna Health and Demographic Surveillance Site (HDSS) in north western Burkina Faso, which has been gathering demographic and epidemiological health information data since 1992. The 59 villages and the town of Nouna that comprise the HDSS have a population of slightly over 100 000 individuals and include one hospital and 13 primary health centres (CSPS).[41]

Our study used longitudinal data from two Burkina Faso waves of the Africa Research, Implementation Science and Education adolescent health study,[42] collected in

the Nouna HDSS in 2017 and 2018. Data were collected from 1644 adolescents aged 12–20 in 2017, based on a stratified random sample of 2544 age-eligible residents in Nouna town and 10 villages.[42 43] A follow-up round was conducted in 2018, attempting to contact all those who participated in 2017; 1291 interviews were completed. In both years, a standardised questionnaire was used to collect self-reported information on sociodemographics, health practices, health outcomes and risk factors. Data were collected by field staff with background in public health, medicine or a related field who had experience in conducting research and had general knowledge about local culture, health issues and the population under study. All study staff received in-depth training at the beginning of the study, covering the topic of research, human research ethics, the study protocol, questionnaire modules, electronic data entry and the procedures for implementing the study, including anthropometric evaluation.

### Patient and public involvement
No patient involved.

### Measures
We used two primary outcomes of youth violence, both captured as count variables: experience of physical attack in the past 12 months; and experience of bullying in the past 30 days (where bullying was defined as physical attacks, threats, insults, frequent nasty teasing, being left out on purpose or having rumours passed about them). We also generated binary measures of any experience for each outcome, in alignment with the World Report of Violence and Health's definitions.[2] Our exposures were age (in completed years) and education (years of full-time education).

We additionally considered a range of time-varying covariates at the individual and household levels (based on Heise framework[20]), plus media use.[44 45] Specifically, our individual-level covariates were: currently in school; marital status (never married vs all else); any work in the past 12 months and sexual behaviour (sexual debut and number of sexual partners). Our household-level covariates were: household size; household wealth quintile; parental vital status; parental support level (16 point scale, converted to the first principal component of the four variables included); parental coresidence (respondent lives with both parents, only mother, only father or lives alone); respondent has their own bedroom. Our media covariates were: any access to television; and frequency of watching television or reading magazines (never, rarely, often, very often). Full variable definitions are provided in online supplemental tables 1 and 2.

### Statistical analyses
We first described our data using frequency and percentages in both waves, including a comparison of those lost to follow-up versus those completing both waves. We then dropped any respondents who were missing data for

the outcome variables, either due to preferring not to respond, not know their answer or where fieldwork errors affected responses, while for some the question was not applicable.

When considering the causal effect of education and age on youth violence, a major concern is unobserved confounding. Given the difficulty of implementing randomised controlled trials, since age is not directly manipulatable and intentional exposure to violence unethical,[46 47] we exploited the panel nature of the data structure to run fixed-effects regression models to remove all time-invariant confounding. We specified our model as:

$$y_{it} = \alpha_i + \beta X_{it} + \gamma Z_{it} + \delta_t + \rho_i \qquad (1)$$

Where $y_{it}$ is youth violence for each individual $i$ at each time point $t$, $X_{it}$ represents our time-varying exposures (education and age), $Z_{it}$ is other time-varying factors for each individual, $\delta_t$ is a period-specific fixed effect to capture all individual-invariant factors and $\rho_i$ are individual-specific fixed-effects which capture all time-invariant factors for each individual, for example, gender, ethnicity, underlying proclivity to violence.

For each outcome (bullying and physical attack), we implemented three linear regression models of the count of reported events, that is., assuming an observation-specific error structure $\epsilon_{it} \sim N(0, \sigma^2)$. We attempted to use Poisson and negative binomial models, that is, modelling $y_{it}$ as count data using a log-link and assuming that the variance of $y_{it}$ is either equal to its mean (Poisson) or its mean plus a dispersion term (negative binomial), however, neither model converged.

Model 1 considered only mean-centred age and years of full-time education. In model 2, we add all time-varying covariates. In model 3, we included interaction terms for gender with age and education to identify any gender-specific effects.

## RESULTS
### Description of sample
At baseline in 2017, 1644 young people were interviewed, of whom 948 (57.7%) were male. By 2018, 21.5% of respondents, comprising 167 (24.0%) females and 186 (19.6%) males, were lost to follow-up, leaving 1291 respondents who participated in both rounds. We dropped 32 individuals (64 data points) who had missing values for the question on bullying, leaving 1258 respondents for the bulling analysis. Similarly, we dropped 14 individuals (28 data points) who did not answer physical attack question to arrive at 1276 respondents. Figure 1 provides a flow chart of how the data were managed. We compared those who were and were not lost to follow-up (online supplemental table 3) and found only one significant difference with those who were retained, those not reinterviewed were less likely to be in school and had lower school attainment in wave 1.

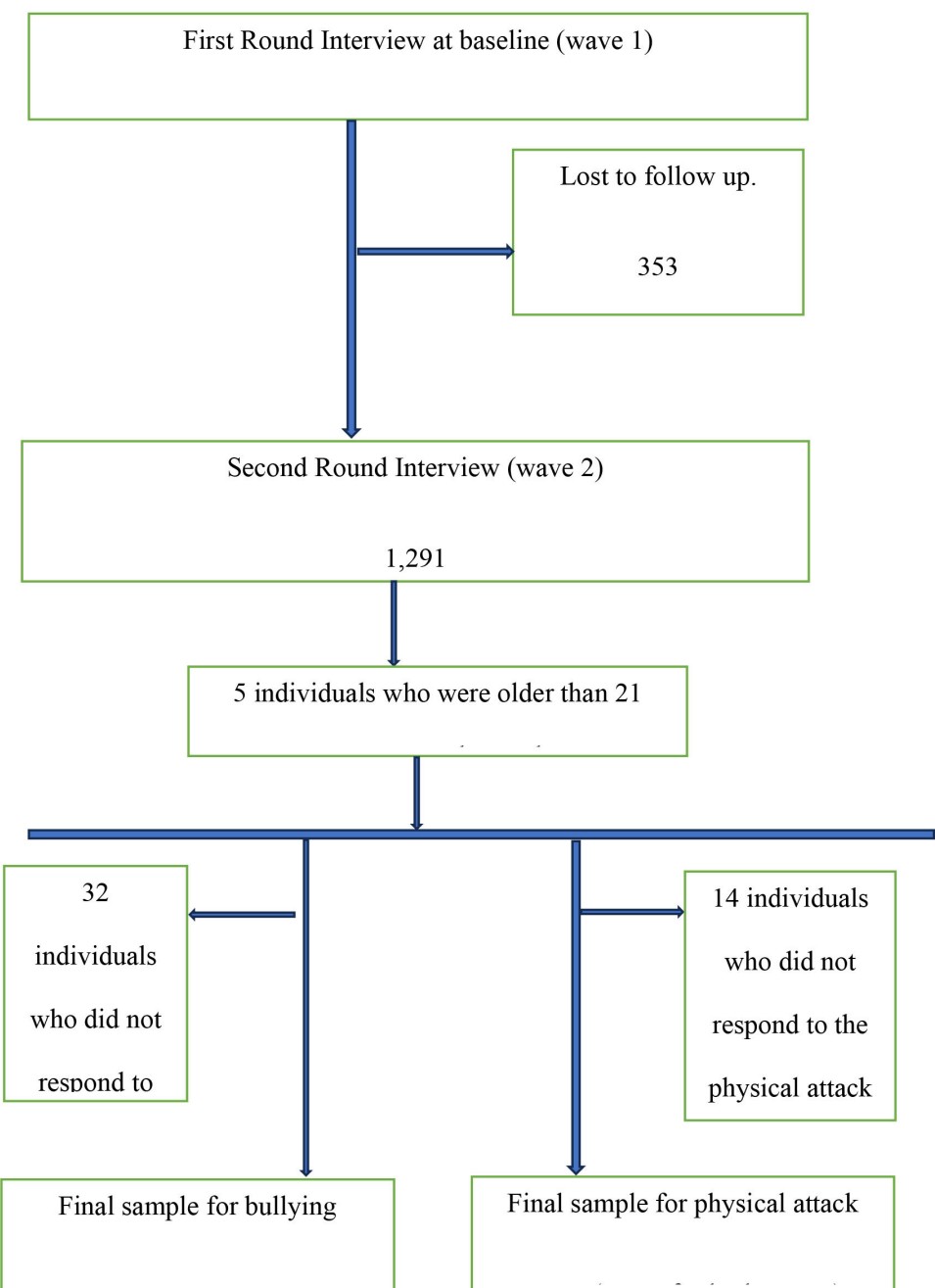

First Round Interview at baseline (wave 1)

Lost to follow up.

353

Second Round Interview (wave 2)

1,291

5 individuals who were older than 21

32 individuals who did not respond to

14 individuals who did not respond to the physical attack

Final sample for bullying

Final sample for physical attack

**Figure 1** Flow chart of sample.

The 1291 respondents were aged 12–20 years in 2017: median 15.5, IQR: 14–18 (see figure 2A). Around half were enrolled in school at each interview: 703 (54.5%) in 2017; 671 (52.0%) in 2018 (see figure 2B).

Around one-quarter of respondents never completed a year of full-time education, while most others had at most attended primary or postprimary level. Over 90% of respondents were single at both interviews, and less than 20% in both 2017 and 2018 reported ever having sexual intercourse (table 1). The proportion of respondents who ever worked fell from over 60% in 2017 to under 42% in 2018. Most respondents had living fathers (over 97%) and mothers (over 91%), however, around one-quarter did not live with their parents. Media access was

mixed: about 20% had no access to television in 2017 but around 15% watched several hours a day (access dropped by 2018); magazine reading was rare. Household wealth was by design evenly distributed across wealth quintiles. Households were large, with a median of 8 or 9 members.

A substantial minority of respondents experienced bullying and physical attacks (table 2). Overall, 189 females (18.9%) and 416 males (27.6%) experienced bullying in the 30 days preceding the interview, while 111 females (10.6%) and 199 males (13.2%) experienced physical attacks in the preceding 12 months. Across both rounds, males experienced both more violence than females: males experienced 416 of 605 (68.8%) unique bullying instances in the past 30 days, and 199 of 310 (64.2%)

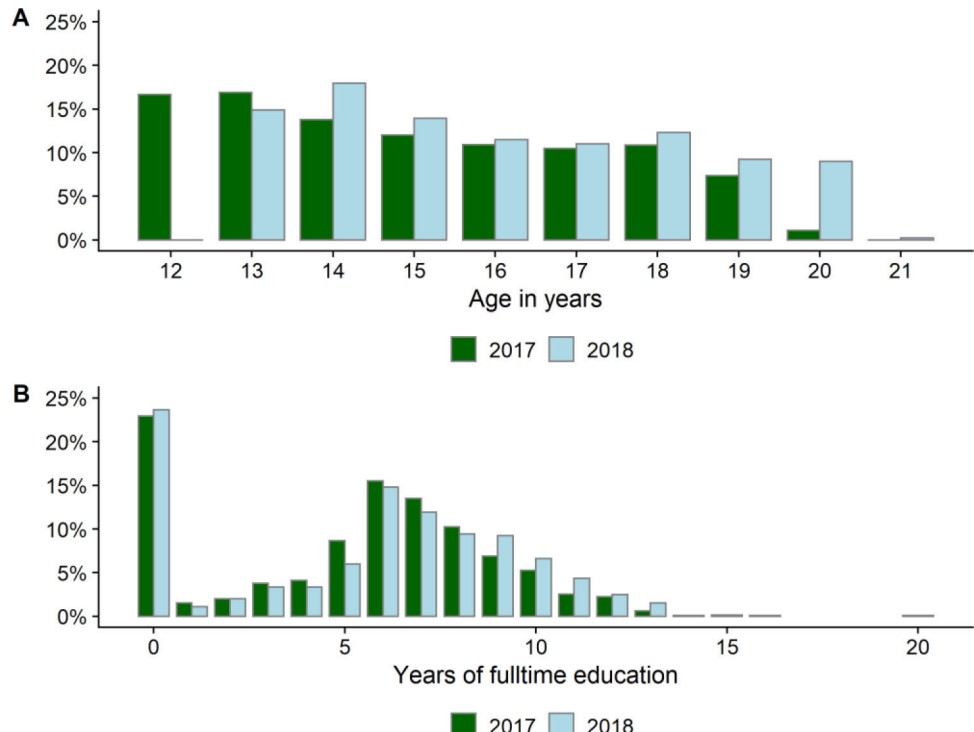

**Figure 2** Exposure distribution among respondents.

unique violence instances in the past 12 months. Bullying experience declined slightly with age—from over 26% among under 15s to 21% among over 18s; physical attacks fell more sharply, especially in early adolescence.

### Fixed-effects analysis

In bivariate fixed-effects regression, older age was associated with non-significantly more bullying (0.14 more bullying experiences per month for each additional year of age, 95% CI –0.12 to 0.39) and associated with significantly fewer physical attacks (–0.19, 95% CI –0.32 to –0.06). More education was associated with significantly more bullying (0.11 bullying experiences per month for each additional year of schooling, 95% CI 0.01 to 0.21), but not with physical attacks (table 3, model 1). Controlling for time-varying potential confounders attenuated the association of age and bullying but otherwise had limited effects on our relationship of interest (table 3, model 2). When we allowed effects to vary by gender (table 3, model 2), associations in all four models were more positive for men than for women, with wider gaps for the impact of age on both outcomes than for education. All these results were independent of a substantial but imprecise negative association between currently being in school and bullying or attacks.

### DISCUSSION

In this study, we employed individual and time fixed-effects models to assess the effects of age and education on violence experience in the form of bullying and physical attacks among adolescents and young adults in a panel study in rural Burkina Faso. We found bullying

experience prevalence in the past 30 days ranging from 20% to 30%, and physical attack experience of 10%–15% in the previous 12 months. While there is little directly comparable data in Burkina Faso, both levels seem concerning, if in line with studies elsewhere.[3 39] In fixed-effects models, we found bullying was associated with more education and weakly with greater age, both effects stronger in males, while physical attacks were associated with younger age (again more strongly for males) but not with education.

Our findings on the effect of age on youth experience of physical attacks are consistent with some observational evidence elsewhere, for example, physical violence from both peers and caregivers falls with age in LAC.[39] Evidence on the causal effect of age on violence is rare, with even cross-sectional studies focused on youth uncommon.[3 11 26] Studies of IPV among women suggest rates are higher in older teenage girls compared with adults,[48 49] but this does not allow within-adolescence comparisons. The faster decline with age that we see for males (from a higher initial level) is important to note: while criminal interpersonal violence appears peaks around age 18 in many settings, our findings and past work suggest that overall frequency of violence experience may in fact decline across teenage years, at least in non-urban settings.[50] In combination, this evidence suggests a shift in violence experience composition for adolescent males that would be worth further investigation.

The implications of this negative association between age and violence experience in adolescence depend on what mechanisms are generating them. First, age might be a proxy for predictors of violence that we have not

**Table 1** Descriptive statistics of independent variables (in per cent)

| | Bullying sample | | Physical attacks sample | |
|---|---|---|---|---|
| | **2017** | **2018** | **2017** | **2018** |
| N | 1253 | | 1271 | |
| Marital status | | | | |
| Single versus all other | 90.7 | 89.6 | 90.8 | 89.6 |
| Living situation | | | | |
| Ever worked versus never | 62.4 | 42.1 | 61.4 | 41.7 |
| Mother is alive | 97.6 | 96.8 | 97.7 | 97.0 |
| Father is alive | 91.7 | 89.7 | 91.9 | 89.8 |
| Lives with mother | 78.0 | 80.1 | 77.7 | 80.3 |
| Lives with father | 74.8 | 76.1 | 74.6 | 76.1 |
| Lives alone | 4.5 | 0.2 | 4.4 | 0.2 |
| Has own bedroom | 18.0 | 19.9 | 17.8 | 19.7 |
| Household wealth quintile | | | | |
| Lowest | 19.0 | 20.8 | 19.8 | 21.0 |
| Second lowest | 19.5 | 20.4 | 20.0 | 20.7 |
| Middle | 22.4 | 19.0 | 22.1 | 19.0 |
| Second highest | 18.4 | 20.1 | 17.9 | 20.2 |
| Highest | 20.8 | 19.6 | 20.2 | 19.1 |
| Education | | | | |
| Currently in school | 54.0 | 51.6 | 54.6 | 52.1 |
| Highest school level | | | | |
| None | 46.1 | 48.4 | 45.5 | 47.9 |
| Primary (1–6) | 20.3 | 10.8 | 20.8 | 11.3 |
| Post-primary (7–10) | 29.9 | 33.6 | 30.1 | 33.7 |
| Secondary (1–3) | 2.6 | 3.9 | 2.6 | 3.9 |
| University | | 0.1 | | 0.1 |
| Not applicable | 1.1 | 3.2 | 1.1 | 3.2 |
| Media use | | | | |
| Has access to television | 80.5 | 70.6 | 80.1 | 70.3 |
| Frequency of watching television | | | | |
| Never | 20.1 | 30.8 | 20.5 | 31.0 |
| Rarely (some hours per month) | 22.8 | 22.8 | 22.7 | 22.8 |
| Often (several hours per week) | 42.1 | 36.2 | 41.8 | 36.1 |
| Very often (several hours per day) | 15.1 | 10.1 | 15.1 | 10.2 |
| Frequency of reading magazines | | | | |
| Never | 90.9 | 87.1 | 91.1 | 87.3 |
| Rarely (some hours per month) | 2.4 | 8.5 | 2.23 | 8.4 |
| Often (several hours per week) | 2.4 | 4.1 | 2.4 | 3.9 |
| Very often (several hours per day) | 4.3 | 0.3 | 4.3 | 0.3 |
| Sexual behaviour | | | | |
| Ever had intercourse | 17.3 | 19.4 | 15.8 | 18.9 |
| No of lifetime sexual partners | | | | |
| 1 | 9.7 | 13.5 | 9.4 | 13.3 |
| 2–7 | 3.8 | 5.2 | 3.8 | 5.3 |
| 8–17 | 0.3 | | 0.3 | |

**Table 1** Continued

|  | Bullying sample | | Physical attacks sample | |
|---|---|---|---|---|
|  | **2017** | **2018** | **2017** | **2018** |
| No response | 2.1 | 1.3 | 2.4 | 0.3 |
| Household size* | 9 (6–12) | 8.5 (7–11) | 8.5 (6–12) | 8 (7–11) |
| Parental support | 7 (4–12) | 10 (7–13) | 7 (4–11) | 10 (7–13) |

*Depict medians and IQRs. Samples are those with non-missing outcome responses for each of the two measures.

captured in this analysis. This might include adolescent autonomy in decision-making, for example, relating to bedtime and the amount and type of television watched, which typically rises with age,[51] : adolescent autonomy was negatively associated with youth violence among US Latino youth.[52] Alternatively, age might be a distal determinant of factors leading more directly to violence experience. For example, older adolescents may be better able to protect themselves against aggressive behaviour from their peers or adults. If this is the case, then structural interventions (at the family, community or national levels) or behaviour change interventions (eg, teaching adolescents how to avoid confrontations) might be beneficial in protecting younger adolescents.[53 54] Further research to understand these causal mechanisms, and thus design effective interventions, will need to include more detailed quantitative data on who perpetrates violence against younger adolescents and qualitative information on how and why such violence comes about.

**Table 2** Distribution of violence experience across observations by age and gender

| Variable | N | Bullying (%) | N | Physical attacks (%) |
|---|---|---|---|---|
| Gender | | | | |
| Female | 996 | 189 (18.9) | 1038 | 111 (10.6) |
| Male | 1510 | 416 (27.6) | 1504 | 199 (13.2) |
| Age | | | | |
| 12 | 19 | 52 (26.1) | 210 | 51 (24.3) |
| 13 | 387 | 100 (25.8) | 398 | 73 (18.3) |
| 14 | 389 | 104 (26.7) | 403 | 65 (16.1) |
| 15 | 327 | 76 (23.2) | 331 | 30 (9.1) |
| 16 | 286 | 67 (23.4) | 288 | 22 (7.6) |
| 17 | 274 | 64 (23.4) | 272 | 27 (9.9) |
| 18 | 297 | 63 (21.2) | 295 | 20 (6.8) |
| 19 | 214 | 43 (20.1) | 212 | 15 (7.1) |
| 20 | 130 | 35 (26.9) | 130 | 7 (5.4) |
| 21 | 3 | 1 (33.3) | 3 | 0 (0.0) |
| Total | 2506 | 605 (23.8) | 2542 | 310 (12.2) |

Each individual is represented twice in this table, once per survey round.

Our finding that bullying increases with age contradicts some past research. Observationally, bullying victimisation rates are higher in younger children than in older adolescents, both in the USA and in sub-Saharan Africa.[55 56] Our results may reflect the stronger control our fixed-effects approach provides against between-individual and temporal confounding, suggesting that the decline in bullying seen elsewhere is a function of factors associated with age, rather than age itself. Our finding of a stronger association for males adds to a mixed literature, aligning with studies from Taiwan and Saskatchwan Canada,[57 58] but in contrast to findings from the USA and Manitoba Canada.[57 59]

We found that education was not associated with violence experience in our setting. A similar null effect of education (grade attainment) was reported for Malawian women aged 19–31 years.[27] A recent meta-analysis of 86 studies in 60 LMICs noted that poor academic performance and weak school attachment were correlated with increased youth violence,[23] in contrast to our null findings. Again, more detail is available for IPV, with a Ugandan study finding that less-educated women were more likely to experience physical IPV—however, this study included women aged 15–49, which makes direct comparison difficult.[60] Other studies have confirmed the protective effect of education on violence in different settings.[48 61] The discrepancy between others' findings and ours may reflect the majority of past studies being cross-sectional, while we were able to use panel data. It may also reflect different exposures, since we considered quantity (years of schooling) rather than quality (performance or attachment).

Past evidence on the effect of education on bullying is mixed, with several studies finding lower bullying among those with more education,[56 57] and few finding the opposite.[59] Our finding of a positive association between education and bullying in both males and females may reflect the a true causal association, or the residual effect of being in school—given the opportunities that this provides for bullying relative to the alternative settings of field-based work or animal herding.

Finally, our analysis covers a population where half of adolescents are not now, and one-quarter never have been, attending school. The role of education in promoting or protecting against violence at the community level may be different in settings where education is not even close

**Table 3** Ordinary least squares models predicting count of violence events

| | Model 1 unadjusted | Model 2 adjusted | Model 3 adjusted and interaction | Interaction test |
|---|---|---|---|---|
| Bullying in the last 30 days (N=2506) | | | | |
| Age in years | 0.14 (–0.12, 0.39) | 0.08 (–0.17, 0.33) | | |
| Female | | | 0.08 (–0.39, 0.24) | |
| Male | | | 0.18 (–0.10, 0.45) | F=2.76, p=0.10 |
| Full-time education in years | 0.11 (0.01, 0.21) | 0.12 (0.02, 0.22) | | |
| Female | | | 0.07 (–0.11, 0.26) | |
| Male | | | 0.14 (0.03, 0.26) | F=0.39, p=0.53 |
| Currently in school | | 0.43 (–1.05, 0.19) | 0.41 (–1.03, 0.21) | |
| Physical attacks in the last 12 months (N=2542) | | | | |
| Age in years | 0.19 (–0.32 to –0.06) | 0.18 (–0.31 to –0.05) | | |
| Female | | | 0.10 (–0.26, 0.05) | |
| Male | | | 0.23 (–0.38 to –0.09) | F=2.74, p=0.10 |
| Full-time education in years | 0.02 (–0.03, 0.07) | 0.04 (–0.01, 0.09) | | |
| Female | | | 0.06 (–0.03, 0.16) | |
| Male | | | 0.03 (–0.03, 0.09) | F=0.41, p=0.52 |
| Currently in school | | 0.27 (–0.59, 0.05) | 0.27 (–0.59, 0.04) | |

The table presents four regression models per column, all showing point estimates and 95% CIs. All models contain fixed-effects for each respondent and interview round. Age centred at 15. Models 2 and 3 are also adjusted for marital status, ever worked, household size, parent alive, living with parents, household wealth, has own bedroom and access to media; full results available in online supplemental table 4.

to universal. Further investigation of why our results looks different from other settings, including qualitative study of social norms surrounding violence across levels of educational attainment, would be instructional.

**Strengths and limitations**

A key strength of our study is the fixed-effects design which robustly removes time-invariant confounding of effect measures.[62] Nevertheless, our study also has potential limitations: while fixed-effects analyses are powerful, they cannot control for unmeasured time-varying confounding and we may have therefore not fully accounted for factors that change rapidly among adolescents, such as increased social media access or social network change. The use of self-reported data may have led to reporting biases, although these would have had to vary differentially over time within respondents in order to bias our fixed-effects analyses. Generalisability from a sample drawn from a single district is always difficult to assess. However, by using a population-based sampling frame our analysis provides results that represent the entire local population and are likely to be broadly applicable in poor, rural settings in Burkina Faso and beyond.

**Conclusion**

A substantial minority of adolescents in rural north-western Burkina Faso report recent experiences of bullying or physical attack. We hypothesised years of education received and age would be associated with violence experience in Burkina Faso. Our findings show the prevalence of these experiences was not significantly associated with years of education received, even within individual respondents, but did fall with age. However, our study was not able to identify mechanisms behind these associations, and we, therefore, recommend a mixed-method study that includes study of household dynamics to move beyond an individualised understanding of violence among adolescents. Such an understanding is central to designing interventions to better protect youth in low-income settings from violence.

**Author affiliations**
[1]Department of Economics, Faculty of Social Science and Arts, Simon Diedong Dombo University of Business and Integrated Development Studies, Wa, Ghana
[2]Centre de Recherche en Sante de Nouna, Nouna, Burkina Faso
[3]Université de Ouagadougou, Ouagadougou, Burkina Faso
[4]Heidelberg Institute of Global Health, Heidelberg University, Heidelberg, Germany
[5]Institute for Global Health, University College London, London, UK
[6]Africa Health Research Institute, KwaZulu-Natal, South Africa
[7]Department of Global Health and Population, Harvard T.H. Chan School of Public Health, Boston, MA, USA
[8]School of Nursing & Public Health, College of Health Sciences, University of KwaZulu-Natal, Durban, South Africa
[9]MRC/Wits Rural Public Health & Health Transitions Research Unit (Agincourt), University of the Witwatersrand, Johannesburg, South Africa

**Contributors** NK: conceptualisation, methodology, formal analysis, writing—original draft. MB: investigation, writing—review and editing, project administration. LO: investigation, writing—review and editing, project administration. AS: investigation, resources, writing—review and editing, supervision. TB: conceptualisation, methodology, writing—review and editing, project administration, supervision, funding acquisition. GH: methodology, data curation, formal analysis, writing—review and editing, supervision, project administration,

guarantor. All authors contributed revisions to the text, approved the final version and agree to be accountable for the work.

**Funding** Funding for the study on which these data are based is provided by the Alexander von Humboldt Foundation through an award to Professor Bärnighausen. GH was supported by a fellowship from the Royal Society and the Wellcome Trust (grant number 210479/Z/18/Z). This research was funded in whole, or in part, by the Wellcome Trust (grant number 210479/Z/18/Z). For the purpose of open access, the authors have applied a CC BY public copyright licence to any Author Accepted Manuscript version arising from this submission.

**Competing interests** None declared.

**Patient and public involvement** Patients and/or the public were not involved in the design, or conduct, or reporting, or dissemination plans of this research.

**Patient consent for publication** Not applicable.

**Ethics approval** The Nouna ARISE study was approved by CRSN's Institutional Ethics Committee (2017/08 and 2018/017) and Heidelberg Medical Faculty's Ethics Commission (S-607/2018). Secondary data analysis was approved by University College London's Research Ethics Committee (15231/005). Prior to the study village elder provided oral assent. Each participants provided written informed consent; for respondents aged under 18, parents or guardians provided written consent alongside the minor's written assent. In cases of illiteracy, a literate witness assisted.

**Provenance and peer review** Not commissioned; externally peer reviewed.

**Data availability statement** Data are available from the corresponding author on reasonable request and after signing a data use agreement.

**Open access** This is an open access article distributed in accordance with the Creative Commons Attribution 4.0 Unported (CC BY 4.0) license, which permits others to copy, redistribute, remix, transform and build upon this work for any purpose, provided the original work is properly cited, a link to the licence is given, and indication of whether changes were made. See: https://creativecommons.org/licenses/by/4.0/.

**ORCID iDs**
Naasegnibe Kuunibe http://orcid.org/0000-0003-1975-3718
Guy Harling http://orcid.org/0000-0001-6604-491X

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
