## [Reviewer comments · BMJ Open]

ARTICLE DETAILS

TITLE (PROVISIONAL)	The effects of education and age on the experience of youth violence in a very low-resource setting: a fixed-effects analysis in rural Burkina Faso.
AUTHORS	Kuunibe, Naasegnibe; Bountogo, Mamadou; Ouermi, Lucienne; Sié, Ali; Bärnighausen, Till; Harling, Guy

VERSION 1 – REVIEW

REVIEWER	Company-Córdoba, Rosalba Universidad Loyola Andalucía, Department of Psychology
REVIEW RETURNED	18-Jan-2023

GENERAL COMMENTS	Firstly, I would like to congratulate the authors for their work on the case of populations that are underrepresented in research. I have some suggestions that could improve your manuscript. Please clarify if reference number 4 included in page 3 is referred to the definition of youth violence or to the affirmation of the consequences of it. In case is not referred to the definition please, include a reference that can explain your concept of this term. In the line 46 of page 4 you indicated that different types of violence is often perpetrated by other females. In case you are not going to focus your research in this particular dyad of violence (female-female), please I would ask to additionally mention if in those countries other types of violence is perpetrated to females by males. I like very much of your study that you focused in the determinants or variables that could explain violence instead of the consequences of it. In line 48 page 4. It would be desirable to extend more the idea that education is a protective factor against violence. You just use one sentence to talk about this concept, and from my point of view this is a core point to explain before you present your study. Which is the definition you have used for bullying?. This should be well clarified. I would recommend basing the statements made in the introduction on a greater number of previous studies. In study design it is necessary to specify which type of professionals were on charge to carry on the interviews to participants.
---

	There are some error messages along the manuscript. Please check. Please confirm that the number 624 is correct in the table 1 In the case “ever worked vs never” or if the correct number is 62,4. It would be recommendable to specify into the title of the table 1 that values correspond to percentages and to maintain into the footnote the values that correspond to medians and interquartile ranges. Page 15 line 9, there is a mistake in the word “research” I suggest that authors include an own opinion or hypothesis for future studies about the reasons behind their results. The discussion section should be improved and worked more deeply. Regarding to strengths and limitations, it is important to refer the gaps in the literature this study covers in order to value the work done in this research.
--	---

REVIEWER	Görlich, Dennis Westfälische Wilhelms-Universität Münster, Institute of Biostatistics and Clinical Research
REVIEW RETURNED	10-May-2023

GENERAL COMMENTS	The authors provided a manuscript presenting their data analysis on the experience of violence (bullying or physical attacks) in a large cohort in Burkina Faso. Overall the results seem convincing and relevant. I nevertheless suggest a major revision due to the following issues. The underlying data is a longitudinal assessment of two time points. The description of methods does not allow to assess within this review how exactly this was handled during data analysis. The used "fixed effect regression" might well be a suitable model for repeated measurements, but the manuscript does not state this clearly [e.g. the foot notes of the supplementary tables mention "fixed effects for each respondent", which might correspond to a "fixed random effect" in a repeated measurement model]. Furthermore, the underlying model for the correlation between repeated measurements is not mentioned. Statistical results and inferences might very well depend on the proper choice, here. Other major comments: Please comment on ethical approval and the type of informed consent collected from the participants. If possible please also reference a study registration. I would suggest to add a flow chart as figure 1 to show how participant data was handled, showing the recruitment waves, reasons for exclusion, sample size of the analyzed data set etc. This might be considered a STROBE flow chart (https://www.strobe-statement.org/). Please provide the completed STROBE checklist as a supplement.
--

	In addition to the major issue describe above please specify your chosen regression model in more detail. The general structure was provided and I assume you used a linear mixed model (i.e. normally distributed residuals). On the other side the authors reported that "Poisson and Negative Binomial models did not converge", which is not further elaborated on. Since both methods would be much more appropriate for count data, please explain your methodology here in more detail. Please elaborate on potential bias and limitation introduced by the responders (who only represent ~20% of the first assessment). The split into "Bullying" and "Physical attacks" Table 1 is a little misleading. I assume that this should indicate that participants who provided this outcome and are "non-missing" in this particular outcome and not that this is the subgroup who experience bullying/physical attacks[which is presented in Table 2]? Minor comments: Please correct the numbering of the affiliations. The first affiliation was numbered with "0". p4,l30: Double word "in" p4,l34: misspelled word "particular" p8,l28: The information on the variable definitions was not linked properly "Full variable definitions are provided in [misspecified reference]" In Table 1, please check result for "Ever worked vs never" where 624 seems not to be a percentage.
--	---

VERSION 1 – AUTHOR RESPONSE

Reviewer 1

Reviewer comments

Firstly, I would like to congratulate the authors for their work on the case of populations that are underrepresented in research. I have some suggestions that could improve your manuscript.

We thank the reviewer.

Please clarify if reference number 4 included in page 3 is referred to the definition of youth violence or to the affirmation of the consequences of it. In case is not referred to the definition please, include a reference that can explain your concept of this term.

We thank the reviewer for this comment. Reference number 4 included in page 3 provides both the definition of youth violence and an affirmation of the consequence of it.

In the line 46 of page 4 you indicated that different types of violence is often perpetrated by other females. In case you are not going to focus your research in this particular dyad of violence (female-female), please I would ask to additionally mention if in those countries other types of violence is perpetrated to females by males.

Certainly, violence does occur from men to women, as well as between women. We originally highlighted female-female violence because this is in addition to the more normative narrative of male-female violence. We have now edited to clarify that both forms of violence occur: AGYW experience all forms of violence (emotional, physical and sexual), often perpetrated both by men and often by other females in domestic and social settings [15-18].

I like very much of your study that you focused in the determinants or variables that could explain violence instead of the consequences of it.

We thank the reviewer for the commendation.

In line 48 page 4. It would be desirable to extend more the idea that education is a protective factor against violence. You just use one sentence to talk about this concept, and from my point of view this is a core point to explain before you present your study.

We are grateful for this suggestion and have expanded as follows: Education is theorized to protect against violence, since more-educated persons are less likely to either perpetrate or experience violence (Behrman, Peterman, & Palermo, 2017; Obradovic-Tomasevic et al., 2019). Evidence shows that women without education were 5.6 times more likely than those with college education to experience intimate partner violence (IPV). Similarly, wives of uneducated men were 1.84 times more likely than those whose husbands had college education to experience IPV. Even at the community level, the likelihood of IPV declined as community male and female literacy increased (after controlling for individual level factors) [28].

Which is the definition you have used for bullying?. This should be well clarified.

We thank the reviewer for the suggestion. We have added the following clarification on page 7 under measurement:

We used two primary outcomes of youth violence, both captured as count variables: experience of physical attack in the past 12 months; and experience of bullying in the past 30 days (where bullying was defined as physical attacks, threats, insults, frequent nasty teasing, being left out on purpose or having rumors passed about them).

I would recommend basing the statements made in the introduction on a greater number of previous studies.

We thank the reviewer for this suggestion but note that we already refer to over 30 publications in the Introduction, many of them either primary analyses or literature reviews.

In study design it is necessary to specify which type of professionals were on charge to carry on the interviews to participants.

We have specified in the type of professionals in the study design as follows: Data were collected by field staff with background in public health, medicine or a related field who had experience in conducting research and had general knowledge about local culture, health issues and the population under study. All study staff received in-depth training at the beginning of the study, covering the topic of research, human research ethics, the study protocol, questionnaire modules, electronic data entry and the procedures for implementing the study, including anthropometric evaluation.

There are some error messages along the manuscript. Please check.

We believe that these arose from links to Supplementary materials that failed when we split the document for submission. Hopefully this issue is now fixed.

Please confirm that the number 624 is correct in the table 1 In the case “ever worked vs never” or if the correct number is 62,4.

We thank the reviewer for spotting this. The number has been corrected as 62.4

It would be recommendable to specify into the title of the table 1 that values correspond to percentages and to maintain into the footnote the values that correspond to medians and interquartile ranges.

We thank the reviewer for the suggestion. We have added (in percent) to the title and took out All values are percent except which are from the footnote.

Page 15 line 9, there is a mistake in the word “research”

We thank the reviewer for spotting this. We have corrected the word “research.”

I suggest that authors include an own opinion or hypothesis for future studies about the reasons behind their results. The discussion section should be improved and worked more deeply.

We thank the reviewer for this suggestion. We have added to the discussion on the “Conclusion” highlighting our opinion on next steps in terms of future studies:

A substantial minority of adolescents in rural north-western Burkina Faso report recent experiences of bullying or physical attack. We hypothesised years of education received and age would be associated with violence experience in Burkina Faso. Our findings show the prevalence of these experiences was not significantly associated with years of education received, even within individual respondents, but did fall with age. However, our study was not able to identify mechanisms behind these associations, and we therefore recommend a mixed method study that includes study of household dynamics to move beyond an individualized understanding of violence amongst adolescents. Such an understanding is vital to important in designing interventions to better protect youth in low-income settings from violence.

Reviewer 2

Reviewer comments

The authors provided a manuscript presenting their data analysis on the experience of violence (bullying or physical attacks) in a large cohort in Burkina Faso. Overall the results seem convincing and relevant. I nevertheless suggest a major revision due to the following issues.

The underlying data is a longitudinal assessment of two time points. The description of methods does not allow to assess within this review how exactly this was handled during data analysis. The used "fixed effect regression" might well be a suitable model for repeated measurements, but the manuscript does not state this clearly [e.g. the foot notes of the supplementary tables mention "fixed effects for each respondent", which might correspond to a "fixed random effect" in a repeated measurement model].

Furthermore, the underlying model for the correlation between repeated measurements is not mentioned. Statistical results and inferences might very well depend on the proper choice, here.

We thank the reviewer for raising this topic. Please note that we have included in the Methods section we explicitly provide the equation used and define all terms in Equation 1:

$$y_{it} = \alpha_i + \beta X_{it} + \gamma Z_{it} + \delta_t + \rho_i \quad (1)$$

Where y_{it} is youth violence for each individual i at each time point t , X_{it} represents our time-varying exposures (education and age), Z_{it} is other time-varying factors for each individual, δ_t is a period-specific fixed effect to capture all individual-invariant factors and ρ_i are individual-specific fixed-effects which capture all time-invariant factors for each individual, e.g., gender, ethnicity, underlying proclivity to violence.

Specifically, the fixed effect is defined as one 'dummy' variable per individual, rather than as a random effect. Once this set of ρ_i terms are included, we do not believe there is the need to add further allowance for a correlation structure between repeated within-person measurements.

Other major comments:

Please comment on ethical approval and the type of informed consent collected from the participants. If possible please also reference a study registration.

We thank the reviewer for their concern regarding ethics. Please note that there is a section at the end of the Methods cover much of this. Since this study was not a trial, there is no additional study registration number.

The Nouna ARISE study was approved by CRSN's Institutional Ethics Committee (2017/08 and 2018/017) and Heidelberg Medical Faculty's Ethics Commission (S-607/2018). Secondary data analysis was approved by University College London's Research Ethics Committee (15231/005). Prior to the study village elder provided oral assent. Each participants provided written informed consent; for respondents aged under 18, parents or guardians provided written consent alongside the minor's written assent. In cases of illiteracy, a literate witness assisted.

I would suggest to add a flow chart as figure 1 to show how participant data was handled, showing the recruitment waves, reasons for exclusion, sample size of the analyzed data set etc. This might be considered a STROBE flow chart (<https://www.strobe-statement.org/>). Please provide the completed STROBE checklist as a supplement.

We have now added a flow chart as Figure 1.

In addition to the major issue describe above please specify your chosen regression model in more detail. The general structure was provided and I assume you used a linear mixed model (i.e. normally distributed residuals). On the other side the authors reported that "Poisson and Negative Binomial models did not converge", which is not further elaborated on. Since both methods would be much more appropriate for count data, please explain your methodology here in more detail.

We appreciate that the existing text was unclear, and have now added text to clarify. We note that while Poisson and Negative Binomial models are a more natural fit, linear regression models are typically robust to moderate non-normality of the outcome variable.

For each outcome (bullying and physical attack), we implemented three linear regression models using count outcome variables (Poisson and Negative Binomial models did not converge). of the count of reported events, i.e., assuming an observation-specific error structure $\epsilon_{it} \sim N(0, \sigma^2)$. We attempted to use Poisson and Negative Binomial models, i.e., modelling y_{it} as count data using a log-link and assuming that the variance of y_{it} is either equal to its mean (Poisson) or its mean plus a dispersion term (Negative Binomial), however neither model converged.

Please elaborate on potential bias and limitation introduced by the responders (who only represent ~20% of the first assessment).

We agree that there is potential for bias to arise from the loss of around 25% of individuals between rounds. We have now built a Supplementary Table to compare those who were and were not reinterviewed at wave 2 and added text to the first paragraph of the Results:

We compared those who were and were not lost to follow up (Supplementary Table 3) and found only one significant difference with those who were retained, those not reinterviewed were less likely to be in school and had lower school attainment in wave 1.

Supplementary Table 3: Difference between those who dropped out in wave 2 and remaining sample.

	participation_w2 = 0	participation_w2 = 1	Test	Statistic	pvalue
	N=353	N=1,291			
Number of physical attacks, last 12m	0 (0-0)	0 (0-0)	Wilcoxon rank-sum	Z= -0.74	0.46
Days bullied, last 30d	0 (0-0)	0 (0-0)	Wilcoxon rank-sum	Z= -1.18	0.24
Gender			Chi-square	Chi2(1)= 4.55	0.033
Female	167 (47.3%)	529 (41.0%)			
Male	186 (52.7%)	762 (59.0%)			
Age (years)	15 (14-18)	15 (13-17)	Wilcoxon rank-sum	Z= 3.64	<0.001
Marital status cat			Chi-square	Chi2(1)= 2.08	0.15
other	41 (11.6%)	117 (9.1%)			
single	312 (88.4%)	1,174 (90.9%)			
Ever worked	222 (62.9%)	786 (60.9%)	Chi-square	Chi2(1)= 0.47	0.49
Mother is alive	342 (96.9%)	1,261 (97.7%)	Chi-square	Chi2(1)= 0.72	0.40

Father is alive	316 (89.5%)	1,186 (91.9%)	Chi-square	Chi2(1)= 1.94	0.16
Lives with Mother	262 (74.2%)	1,005 (77.8%)	Chi-square	Chi2(1)= 2.06	0.15
Lives with father	245 (69.4%)	965 (74.7%)	Chi-square	Chi2(1)= 4.07	0.044
Lives alone	23 (6.5%)	56 (4.3%)	Chi-square	Chi2(1)= 2.87	0.090
Has own bedroom	58 (16.4%)	228 (17.7%)	Chi-square	Chi2(1)= 0.29	0.59
Wealth quintile			Chi-square	Chi2(4)= 1.12	0.89
1	76 (21.5%)	255 (19.8%)			
2	74 (21.0%)	257 (19.9%)			
3	73 (20.7%)	282 (21.8%)			
4	64 (18.1%)	235 (18.2%)			
5	66 (18.7%)	262 (20.3%)			
Currently in school	119 (33.7%)	703 (54.5%)	Chi-square	Chi2(1)= 47.71	<0.001
Current school level			Chi-square	Chi2(4)= 53.56	<0.001
None	234 (67.2%)	589 (46.2%)			
Primary (1-6)	40 (11.5%)	268 (21.0%)			
Post Primary (7-10)	66 (19.0%)	386 (30.3%)			
Secondary (1-3)	7 (2.0%)	33 (2.6%)			
Technical/Vocational	1 (0.3%)	0 (0.0%)			
Has access to TV	295 (83.6%)	1,035 (80.2%)	Chi-square	Chi2(1)= 2.07	0.15
Frequency of watching TV			Chi-square	Chi2(3)= 2.69	0.44
Never	59 (16.8%)	263 (20.4%)			
Rarely (some hours per month)	88 (25.0%)	290 (22.5%)			
Often (several hours per week)	151 (42.9%)	540 (41.9%)			
Very often (several hours per day)	54 (15.3%)	197 (15.3%)			
Frequency of reading magazines			Chi-square	Chi2(3)= 6.12	0.11
Never	334 (94.6%)	1,177 (91.2%)			
Rarely (some hours per month)	8 (2.3%)	30 (2.3%)			
Often (several hours per week)	5 (1.4%)	30 (2.3%)			
Very often (several hours per day)	6 (1.7%)	54 (4.2%)			

Ever had intercourse?	55 (16.8%)	202 (17.0%)	Chi-square	Chi2(1)= 0.01	0.93
Number of sex partners, lifetime			Chi-square	Chi2(11)= 6.01	0.87
0	298 (86.4%)	1,087 (86.3%)			
1	29 (8.4%)	121 (9.6%)			
2	9 (2.6%)	25 (2.0%)			
3	5 (1.4%)	14 (1.1%)			
4	1 (0.3%)	3 (0.2%)			
5	0 (0.0%)	4 (0.3%)			
6	1 (0.3%)	1 (0.1%)			
7	0 (0.0%)	1 (0.1%)			
8	1 (0.3%)	1 (0.1%)			
9	0 (0.0%)	1 (0.1%)			
10	1 (0.3%)	1 (0.1%)			
17	0 (0.0%)	1 (0.1%)			
Scores for component 1	-.2998273 (- 1.694384- .8893526)	-.6083845 (- 1.707631- .9265742)	Wilcoxon rank-sum	Z= 0.57	0.57
Parental_support	2 (1-2.75)	1.75 (1-2.75)	Wilcoxon rank-sum	Z= 0.71	0.48

Data are presented as median (IQR) for continuous measures, and n (%) for categorical measures.

The split into "Bullying" and "Physical attacks" Table 1 is a little misleading. I assume that this should indicate that participants who provided this outcome and are "non-missing" in this particular outcome and not that this is the subgroup who experience bullying/physical attacks [which is presented in Table 2]?

We thank the reviewer for this comment, as they suggest the table refers to those who were in the relevant samples. We have clarified this in the footnote to the table:

Samples are those with non-missing outcome responses for each of the two measures.

Minor comments:

Please correct the numbering of the affiliations. The first affiliation was numbered with "0".

We have re-entered this value, which hopefully solves the problem.

p4,l30: Double word "in"

The other "in" has been deleted

p4,l34: misspelled word "particular"

The word is corrected as particular

p8,l28: The information on the variable definitions was not linked properly "Full variable definitions are provided in [misspecified reference]"

This issue arose from the separation of the supplementary material from the main manuscript and has now been corrected.

In Table 1, please check result for "Ever worked vs never" where 624 seems not to be a percentage.

We thank the reviewer for spotting this. The figure has been corrected to 62.4